# A Comprehensive Review of Elbow Exoskeletons: Classification by Structure, Actuation, and Sensing Technologies

**DOI:** 10.3390/s25144263

**Published:** 2025-07-09

**Authors:** Callista Shekar Ayu Supriyono, Mihai Dragusanu, Monica Malvezzi

**Affiliations:** Department of Information Engineering and Mathematics, University of Siena, 53100 Siena, Italy; c.supriyono@student.unisi.it (C.S.A.S.); monica.malvezzi@unisi.it (M.M.)

**Keywords:** elbow exoskeleton, assistive, rehabilitation, augmentation, sensor, actuator

## Abstract

The development of wearable robotic exoskeletons has seen rapid progress in recent years, driven by the growing need for technologies that support motor rehabilitation, assist individuals with physical impairments, and enhance human capabilities in both clinical and everyday contexts. Within this field, elbow exoskeletons have emerged as a key focus due to the joint’s essential role in upper limb functionality and its frequent impairment following neurological injuries such as stroke. With increasing research activity, there is a strong interest in evaluating these systems not only from a technical perspective but also in terms of user comfort, adaptability, and clinical relevance. This review investigates recent advancements in elbow exoskeleton technology, evaluating their effectiveness and identifying key design challenges and limitations. Devices are categorized based on three main criteria: mechanical structure (rigid, soft, or hybrid), actuation method, and sensing technologies. Additionally, the review classifies systems by their supported range of motion, flexion–extension, supination–pronation, or both. Through a systematic analysis of these features, the paper highlights current design trends, common trade-offs, and research gaps, aiming to guide the development of more practical, effective, and accessible elbow exoskeletons.

## 1. Introduction

The human upper limb, particularly the elbow joint, plays a critical role in performing a wide range of daily activities, from simple tasks like eating to complex functions in professional settings [1]. However, injuries, aging, neurological conditions, or physical impairments can severely affect the functionality of the elbow, limiting mobility and independence [2,3,4]. As the demand for assistive technologies grows, elbow exoskeletons have emerged as a promising solution for both the rehabilitation and augmentation of human motion. These wearable systems are specifically designed to assist users in performing elbow-related movements, supporting joint function and facilitating natural motion. Based on the natural biomechanics of the human elbow, these systems enable the mechanical transfer of power to produce two primary types of joint motions: flexion–extension and supination–pronation.

An exoskeleton is a wearable device designed to support and assist the human body by moving in harmony with its natural motions [5,6]. Numerous models of exoskeletons have been created and developed to serve various purposes, from rehabilitation to industrial applications. Elbow exoskeletons, as a subset of this technology, are created to provide support during elbow motions, either assisting with movement or aiding in rehabilitation processes.

Elbow exoskeletons are wearable devices including actuators, sensors, and control systems that facilitate smooth interaction between the user and the device. The actuators actuate or support natural limbs’ and joints’ movement, while sensors monitor system status, for example, joint angles, muscle activity, forces, and other key variables. Control algorithms then interpret this sensor feedback to provide the appropriate level of assistance or, in rehabilitation contexts, controlled resistance depending on the user’s needs. As these devices evolve, their design has become increasingly sophisticated, incorporating advances in robotics, biomechatronics, and sensor technologies. According to Tiboni and colleagues’ recent review [7], the elbow joint is the most considered in the design of devices for upper limb actuation.

This review paper aims to provide a comprehensive overview of the current state of elbow exoskeletons, with a focus on their design, actuation systems, sensor technologies, control strategies, and evaluation methods. We will examine the major actuation types, including pneumatic, hydraulic, cable-based, and electric systems, and their suitability for various applications. Additionally, we will explore the key sensors used for feedback and control, such as force sensors, torque sensors, EMG sensors, and IMUs. Finally, the paper will discuss the challenges faced in their development and the emerging trends in the field. The review concludes with a discussion on future directions, including the integration of artificial intelligence, soft robotics, and the commercialization of these technologies for broader use in clinical and industrial settings.

## 2. Methodology

To analyze the development trends of elbow exoskeleton technologies, we conducted a targeted literature search using Google Scholar (Google LLC, California, USA) as the sole database. We selected Google Scholar as the sole database for our literature search due to its comprehensive coverage of scholarly publications across a wide range of disciplines, including engineering, robotics, rehabilitation, and biomedical sciences. The initial search included publications related to upper-limb exoskeletons to provide broader context and ensure that no relevant elbow-focused research was missed. However, for the purposes of this study, we included only papers that specifically addressed elbow exoskeletons for our main analysis.

We used a combination of keywords including “elbow exoskeleton”, “upper-limb exoskeleton”, “rehabilitation”, “assistive”, “augmentation”, “sensor”, and “actuator” to identify relevant peer-reviewed journal articles and conference papers. Articles published between 2010 and 2025 were prioritized to capture both early developments and recent innovations. After removing duplicates and filtering out unrelated studies, we screened the remaining articles by title, abstract, and full text to ensure technical relevance and a specific focus on elbow assistance. The inclusion criteria emphasized academic rigor and topic relevance, leading us to select only peer-reviewed journal articles and conference papers for further analysis.

In total, 67 research papers were selected based on their technical depth, relevance, and contribution to the understanding of elbow exoskeleton design and development. From this pool, we identified and analyzed 17 individual elbow exoskeleton devices, chosen for their availability of detailed technical specifications and representation of diverse design approaches. Each device was evaluated across several key dimensions, including mechanical structure, degrees of freedom (DOFs), type of actuators, and the roles of integrated sensors.

An analysis of publication years revealed a clear trend: early research (2010–2015) focused primarily on mechanical design and proof-of-concept systems. From 2016 onward, the literature shows a growing emphasis on soft robotics, user adaptability, and wearable assistive systems. More recent studies increasingly address real-world applications, integration with sensors, and user-centered evaluation, indicating the field’s gradual shift from laboratory prototypes toward clinical and commercial readiness.

Although many new exoskeleton systems have been introduced in recent years, our analysis was limited to those that explicitly target the elbow joint. This focused approach enabled us to identify and evaluate trends unique to this subfield within the broader landscape of wearable robotics. The reviewed literature spans the last 15 years, allowing us to trace the evolution of elbow exoskeleton technologies and gain insights into emerging research directions, design maturation, and the transition toward more refined, market-ready solutions. By analyzing how these devices have changed in structure and purpose over time, we aim to offer both a comparative assessment of current technologies and a contextual understanding of the field’s development.

## 3. Anatomy of Human Elbow

The human body is divided into three main sections: the upper limbs, the trunk, and the lower limbs. The upper limb itself consists of three sections: the upper arm, forearm, and hand. Within these sections, we find important structures crucial for movement, such as the shoulder complex, elbow complex, wrist joint, and fingers. These structures work in harmony to provide the dexterity and flexibility necessary for complex tasks. The human upper limb features seven degrees of freedom (DOFs): three in the shoulder, two in the elbow, and two in the wrist. As noted by Gull and colleagues, most studies model the elbow as having only one degree of freedom, typically for flexion–extension. However, the forearm enables a second rotational movement supination–pronation which involves both the proximal and distal radioulnar joints, though it is often functionally associated with the elbow. While research often focuses on flexion–extension due to its simpler implementation, supination–pronation is equally important, particularly for tasks like eating and drinking, which require the rotation of the forearm [8].

### 3.1. Bones and Joint Structure

The elbow is a complex joint system composed of three bones: the humerus, radius, and ulna, which interact through three primary articulations: the humeroulnar (HU), humeroradial (HR), and proximal radioulnar (RU) joints. These interconnected joints enable two active degrees of freedom (DOFs): flexion–extension and pronation–supination, along with one passive DOF, varus–valgus (adduction–abduction) motion [9]. The humeroulnar joint primarily supports the hinge-like flexion and extension, while pronation and supination—the rotational movements of the forearm—are facilitated by the humeroradial and proximal radioulnar joints, rather than by the radius alone.

The distal humerus features the trochlea (medial side) and capitellum (lateral side), which articulate with the ulna and radius, respectively. This anatomical arrangement permits complex motion patterns involving rotation, sliding, and rolling across multiple planes. Consequently, describing the elbow as a simple hinge joint is a simplification; it is more accurately characterized as a multi-joint structure with coordinated biomechanical functions, critical for executing everyday activities requiring both strength and precision [10].

### 3.2. Muscles and Tendons

Key muscles involved in elbow movement include the biceps brachii, triceps brachii, and brachialis, among others. These muscles work together to control the motion of the elbow. The first one is the biceps brachii; responsible for elbow flexion, the biceps help bend the arm at the elbow joint. The second one is the triceps brachii; the triceps oppose the action of the biceps and are responsible for elbow extension, straightening the arm. Then, the third one is the brachialis, a muscle located beneath the biceps, which assists in flexion of the elbow, particularly when the forearm is in a pronated position. Additional muscles, such as the pronator teres and supinator muscles, play key roles in forearm rotation, supination, and pronation [11].

### 3.3. Ligaments

The stability of the elbow joint is maintained by several key ligaments that counteract specific mechanical stresses during movement. The Ulnar Collateral Ligament (UCL) provides medial support by resisting valgus stress, which is the outward angulation of the forearm relative to the upper arm, a force commonly encountered during overhead throwing motions. On the lateral side, the Radial Collateral Ligament (RCL) resists varus stress, or inward angulation, helping to stabilize the elbow against forces that would push it outward. Additionally, the Annular Ligament encircles the head of the radius and holds it in place against the ulna, allowing smooth rotational movement during supination and pronation [12]. These ligaments play a critical role in maintaining joint integrity during both linear and rotational activities.

### 3.4. Movement, Range of Motion, and Force

The primary movement of the elbow joint is flexion and extension, which occurs along the sagittal plane and is primarily facilitated by the humeroulnar and humeroradial joints. The standard anatomical reference for elbow flexion begins at 0°, representing full extension where the arm is straightened alongside the body. However, as shown in Figure 1, some individuals may exhibit slight hyperextension, up to 11° beyond neutral, due to natural joint variability. The normal range of flexion extends up to approximately 154°, enabling common activities such as lifting and reaching [10].

In addition to its hinge-like motion, the elbow joint complex comprising the humeroulnar, humeroradial, and proximal radioulnar joints supports pronation and supination, which are rotational movements of the forearm [9]. These occur between the radius and ulna, specifically at the proximal and distal radioulnar joints, and are crucial for tasks such as turning a doorknob or holding a cup. The typical range for pronation is 0° to 85°, while supination spans 0° to 104° [10]. Although biomechanically associated with the forearm, these motions are functionally coupled with elbow movement and are often considered in the design of elbow exoskeletons.

Regarding torque requirements, pronation and supination typically demand about 0.06 Nm [13], whereas flexion and extension require significantly more, approximately 2.7 Nm [14], highlighting the mechanical demands placed on assistive systems for the elbow. The range of motion (ROM) of the elbow joint is summarized in Table 1.

## 4. Elbow Exoskeleton Application

The concept of exoskeletons was first developed in the late 1960s for military purposes. The first successful prototype introduced in the 1960s was called Hardiman. At that time, the actuator system used was a hydraulic actuator, designed to provide strength assistance to the user [15]. Since this initial introduction, significant developments have been made in the design of elbow exoskeletons due to their potential, not only in military applications but also in improving users’ quality of life and serving various other purposes, such as in industrial applications.

Elbow exoskeletons are commonly found in three main scenarios: assistive, rehabilitation, and augmentation [16].

### 4.1. Assistive Application

According to the World Health Organization (WHO), approximately 16.5 percent of the global population suffers from motor impairments, limiting their ability to engage in physical activities. To address this, lightweight and comfortable elbow exoskeletons have been designed for daily use as an assistive device [17]. These devices are also frequently used by elderly individuals with reduced physical strength to ease their daily routines [18]. In these scenarios, elbow exoskeletons serve as assistive devices, helping users perform daily activities more easily, such as lifting objects, reaching, or carrying groceries. The primary goal is to improve the quality of life for individuals with motor disabilities by providing them with the extra strength and stability they need to perform routine tasks with less effort.

### 4.2. Rehabilitation Application

In addition to being assistive devices, elbow exoskeletons can be used in rehabilitation. One example is in the case of stroke. Stroke, as we know, is one of the most dangerous illnesses and the second-leading cause of death worldwide. According to WHO, 15 million people suffer from stroke each year [19]. However, stroke is still a condition that can be managed and treated if immediate action is taken, and today, many stroke survivors can be found. Despite this, stroke remains a leading cause of upper limb impairment, with many survivors losing the ability to move their arms. For this reason, several elbow exoskeletons have been specifically designed to assist therapists in performing rehabilitative exercises, helping survivors regain mobility [20]. These exoskeletons are equipped with sensors and actuators that allow for controlled, repetitive motion, aiding in neuromuscular recovery. By using exoskeletons in therapy, patients can receive targeted assistance to restore motion, reduce the effects of muscle atrophy, and improve motor control, all while ensuring proper biomechanics during the recovery process [21].

### 4.3. Augmentation Application

Beyond the healthcare applications mentioned earlier, elbow exoskeletons also play an important role in the augmentation field. Building on the original idea first implemented in Hardiman [8], and with technological advancements, elbow exoskeletons are increasingly being used for augmentation purposes, such as assisting industrial workers with lifting and carrying heavy loads or aiding soldiers in military operations. This not only provides additional strength to the user but also protects them from both minor and severe injuries. Minor injuries, such as muscle fatigue, tendon strain, or joint discomfort, commonly arise from repetitive or sustained physical tasks. More severe injuries including musculoskeletal disorders, ligament tears, or spinal compression can result from improper lifting techniques, overexertion, or carrying heavy loads over time. Elbow exoskeletons help mitigate these risks by redistributing mechanical loads, limiting excessive joint movement, and supporting proper posture during strenuous activities [22,23,24]. In industrial settings, for example, workers often face physically demanding tasks that require heavy lifting or repetitive movements, leading to long-term physical strain. Elbow exoskeletons can mitigate the risk of injury by supporting the arm during lifting and improving endurance. Similarly, in military applications, exoskeletons are designed to assist soldiers by supporting load bearing tasks and reducing physical strain, which can help them to carry heavier equipment for extended periods and mitigate fatigue [25]. These devices help improve the overall performance of users in environments where physical exertion is high, making them valuable tools for a wide range of industries and sectors [16,26].

## 5. Elbow Exoskeleton Classification

Elbow exoskeletons can be classified into different types based on their mechanical structure and the nature of the assistance they provide, as shown in Figure 2. These classifications highlight the fundamental design differences and the types of assistance these devices offer to the user.

### 5.1. Rigid, Soft, and Hybrid Elbow Exoskeleton

Based on their mechanical structure, exoskeletons can be categorized into rigid, soft, or even hybrid exoskeletons [27].

Rigid exoskeletons provide strong structural support and precise control but tend to be heavier and more expensive due to material choices. These exoskeletons are often made from PLA, metal, carbon fiber, or other strong materials that offer higher levels of force output and support. They are designed to assist with tasks requiring greater strength and stability, making them ideal for rehabilitation and augmentation applications [1].

While rigid exoskeletons give strong support, soft exoskeletons prioritize comfort and flexibility. They are made from lighter, flexible materials like textiles or polymers, which are easier to wear and better integrated into the user’s daily activities. These exoskeletons provide less mechanical force but offer greater adaptability to the human body, making them especially beneficial for assistive applications. Soft exoskeletons are generally lightweight, breathable, and more comfortable for prolonged use [28,29,30,31].

On the other hand, a hybrid exoskeleton is designed to assist human movement by providing support during activities while maintaining a balance between flexibility and structural stability. Unlike fully soft exoskeletons, which use soft materials (e.g., textile-based materials) for maximum comfortability, hybrid exoskeletons usually use a hard material for their structural support and use a soft material actuator for their actuation system [32,33,34,35].

### 5.2. Active and Passive Elbow Exoskeleton

In addition to the rigid–soft classification, exoskeletons can be categorized into active and passive systems. Active exoskeletons use powered actuators to actively assist the user’s movements. These devices provide greater control and allow for precise, adjustable support, making them ideal for tasks that require significant strength or precision, such as lifting heavy objects or performing rehabilitation exercises. Active systems are commonly used in rehabilitation and augmentation settings, where user performance and strength are enhanced through powered assistance. In contrast, passive exoskeletons rely on springs, damping mechanisms, or other passive elements to assist motion. These systems do not have powered actuators and instead provide support through mechanical advantage, allowing users to move more easily without generating additional power. Passive exoskeletons are typically lighter, less expensive, and simpler in design. They are ideal for users who need less intensive support or for use in less physically demanding activities [36,37,38].

## 6. Elbow Exoskeleton Actuation System

In an active elbow exoskeleton, the actuator serves as the core driving element that generates motion and force to assist, augment, or rehabilitate the elbow joint.

Its primary function is to mimic or support the natural flexion and extension of the elbow. The choice of actuator significantly influences the exoskeleton’s performance, comfort, and suitability for specific applications [7]. There are different types of actuators that have been used in exoskeletons’ actuation: pneumatic, hydraulic, cable-based, electric, shape memory alloy, and Variable Stiffness Actuators. Specifically for elbow exoskeletons, hydraulic and shape memory alloy actuators are less frequent. For the sake of completeness, however, we included them in the description.

### 6.1. Pneumatic Actuator

Pneumatic actuators operate using compressed air to create movement and are known for their lightweight structure and rapid response. In elbow exoskeletons, this makes them particularly appealing for assistive settings where safety and compliance with the user’s natural motion are important. Since air is compressible, pneumatic systems are generally softer and more forgiving, reducing the risk of injury. However, their reliance on external air sources like compressors or tanks limits their portability, and precise motion control is difficult due to the compressibility of air [39].

### 6.2. Hydraulic Actuator

Hydraulic actuators use pressurized fluids to produce high levels of force, offering exceptional power density and fine-grained control. However, their main drawbacks are their size, weight, and the complexity of fluid systems, which require pumps, reservoirs, and hoses. These systems can also leak, creating safety and maintenance issues. Due to these limitations, hydraulics are not ideal for mobile or wearable applications, but they shine in stationary, heavy-duty use cases [40].

### 6.3. Cable-Based

Cable-based actuators, such as Bowden cable systems or Twisted String Actuators (TSAs), use remotely placed motors to transmit force via flexible cables or strings, allowing the joint itself to remain lightweight and minimally encumbered. In TSA systems, a motor twists a pair of strings, causing them to shorten and generate linear motion that can be converted into joint rotation, making the setup compact and mechanically simple [41]. These systems are ideal for soft or wearable exoskeletons where low joint weight and flexibility are critical, such as in assistive daily-use or rehabilitation devices. However, they often suffer from mechanical losses due to friction, cable stretch, and hysteresis, especially in Bowden cables, and they require periodic maintenance to maintain performance [29]. Despite these drawbacks, their ability to offload weight from the joint and conform to the human body makes them a popular choice in ergonomic and user-friendly exoskeleton designs.

### 6.4. Electric Actuator

Electric actuators, like servo or brushless DC motors, convert electrical energy into mechanical motion and offer high precision, control, and reliability. They are commonly used in portable and battery-powered exoskeletons for both assistive and rehabilitative purposes, thanks to their ease of integration with digital control systems. However, delivering high torque directly at the elbow often requires motors that are relatively heavy and may need cooling solutions to prevent overheating. Despite these trade-offs, electric actuators are favored for applications that demand smooth, precise, and responsive movement in a compact and mobile format [42].

### 6.5. Shape Memory Alloy

Shape Memory Alloy (SMA) actuators are smart materials that can return to their original shape when heated. They work by changing shape in response to temperature, which creates movement. This makes them useful for small, lightweight, and quiet devices, like wearable exoskeletons. SMAs are especially good for soft and compact designs because they do not need motors or gears. However, they can be slow to cool down and may not be strong enough for heavy tasks [4,43].

### 6.6. Variable Stiffness Actuator

Variable Stiffness Actuators (VSAs) are specialized robotic actuators that can modify their mechanical stiffness independently from their position or torque control. Unlike conventional actuators with fixed stiffness, VSAs achieve stiffness changes through their mechanical design, avoiding the need for complex control loops and reducing the chance of instability. This feature enables rehabilitation devices using VSAs to adjust the level of assistance based on the patient’s condition and gradually intensify therapy. The ability to vary stiffness improves the robot’s flexibility, energy use, and safety during interaction with humans, making VSAs particularly beneficial in applications like rehabilitation robotics and prosthetics, where mimicking the natural joint stiffness can enhance motor function recovery and user comfort [44,45,46].

### 6.7. Actuators Comparison

Table 2 compares common actuator types used in elbow exoskeletons across five key performance categories: power density, control precision, weight, portability, and compliance. Power density reflects the actuator’s force output relative to size and weight, crucial for wearable applications. Control precision measures how accurately the actuator can replicate natural elbow movements, essential for smooth and effective assistance. Weight and portability impact user comfort and device usability in mobile scenarios, with lighter and more portable actuators being preferable, like TSAs and SMAs. Compliance indicates how safely and comfortably the actuator interacts with the human body, with pneumatic, TSA, SMA, and VSA actuators offering higher compliance due to their inherent material properties or design flexibility. This comparison highlights the trade-offs involved in selecting actuators, where pneumatic and SMA actuators excel in compliance but may lack in power or control speed, while an electric actuator provides higher power and precision but varies in weight and compliance. Meanwhile, TSAs, Bowden cable-driven actuators, and VSAs aim to integrate the best of both worlds in terms of precision, adaptability, and safety.

## 7. Elbow Exoskeleton Sensor Technologies

In an elbow exoskeleton, sensors play a crucial role in enabling intelligent interaction between the user and the device. They provide real-time data that allows the system to interpret user intention, monitor movement, control actuators, and ensure safety. Depending on the purpose of the exoskeleton, be it assistive, rehabilitative, or industrial, different types of sensors are integrated to gather information on muscle activity, joint motion, external forces, and environmental conditions. Among the commonly used sensors in elbow exoskeletons are force sensors, torque sensors, electromyography (EMG) sensors, and inertial measurement units (IMUs). Each of these serves a specific function in enhancing the responsiveness, adaptability, and effectiveness of the exoskeleton system.

### 7.1. Force Sensor

Force sensors are used to measure the amount of linear force being applied, either by the user or the exoskeleton itself. In elbow exoskeletons, load cells are often embedded at the attachment interfaces such as straps or rigid frame connections to monitor how much force is being exerted during movement or assistance [7]. This information is essential for modulating the actuator’s output in real time, ensuring that the device provides the appropriate level of support or resistance. Force sensors also contribute to user safety by detecting unexpected loads that could indicate misuse or a malfunction.

### 7.2. Torque Sensor

Torque sensors provide measurements of rotational force at the joint. These sensors are particularly important in robotic exoskeletons where the control system needs to know how much torque the actuator is delivering or how much torque is being applied by the user [47]. By integrating torque sensors at the joint or motor, the system can precisely balance its assistance or resistance, leading to smoother, more natural movements.

### 7.3. EMG Sensor

EMG sensors detect the electrical activity produced by muscle contractions. Surface EMG sensors, which are non-invasive and placed on the skin, are commonly used on the biceps and triceps in elbow exoskeletons [32,33,34]. These sensors allow the system to interpret the user’s muscle activation patterns, effectively reading their intention to move. This capability is particularly valuable in assistive and rehabilitative devices, where user-driven control enhances engagement and effectiveness.

### 7.4. IMU Sensor

IMUs combine accelerometers, gyroscopes, and sometimes magnetometers to measure orientation, acceleration, and angular velocity [48]. IMUs are lightweight and compact, making them ideal for wearable systems. They are particularly useful in scenarios where continuous motion monitoring is required, such as during rehabilitation exercises or physical assistance tasks.

### 7.5. Sensor Comparison

The sensor comparison (Table 3) categorizes each sensor type based on the quantity it measures such as force, torque, muscle activity, or motion and outlines its practical strengths and limitations. For instance, force and torque sensors offer accurate mechanical feedback but may add weight or require careful calibration. EMG sensors enable user-intent detection through muscle signals, though they are prone to signal noise. IMUs are lightweight and ideal for tracking motion, yet can suffer from drift over time. The Common Applications row shows how these characteristics align with specific use cases, such as rehabilitation, motion tracking, or adaptive control, providing a clear guide for selecting appropriate sensors based on system goals.

## 8. Existing Technologies of Elbow Exoskeleton

The development of elbow exoskeletons has advanced significantly, resulting in a variety of designs that differ in actuator mechanisms, sensor integration, degrees of freedom, and portability. Each configuration serves unique rehabilitation and assistive purposes. Table 4 presents a comprehensive list of elbow exoskeletons that are discussed in this paper, including their key features and specifications. Corresponding images of the devices are provided in Figure 3 to support a visual understanding of the designs.

**Figure 3 sensors-25-04263-f003:**
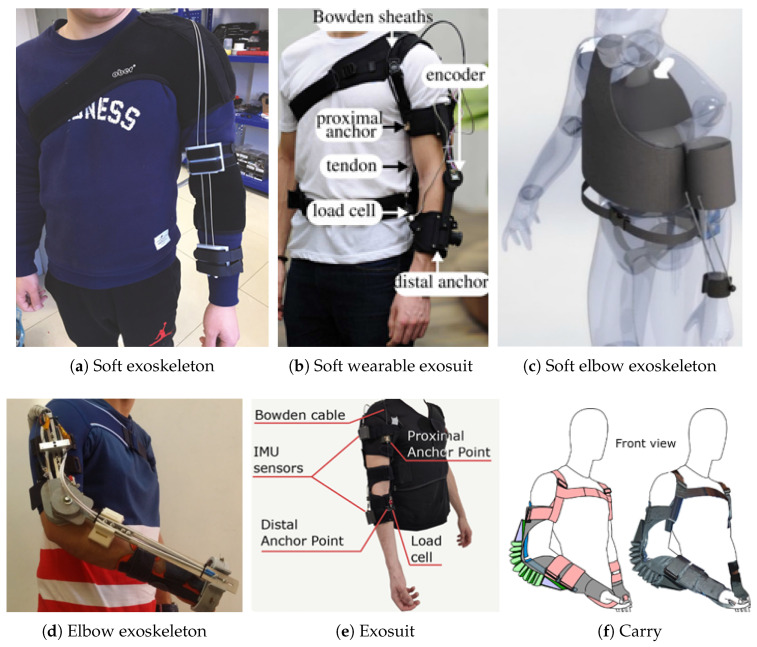
Prototypes of the elbow exoskeletons. (**a**) Soft exoskeleton actuated by a cable-driven mechanism (Bowden cable), incorporating a force sensor for feedback control [29]. (**b**) The soft wearable exosuit utilizing a cable-driven actuator (Bowden cable), also integrated with a force sensor for motion assistance [30]. (**c**) A soft elbow exoskeleton employing a motor tendon actuator, equipped with an infrared sensor, capable of both flexion–extension and supination–pronation movements [31]. (**d**) An elbow exoskeleton actuated by shape memory alloy (SMA) wires, supporting both flexion–extension and supination–pronation [49]. (**e**) The lightweight exosuit with an inertial measurement unit (IMU) sensor, actuated via a Bowden cable for motion assistance [6]. (**f**) “Carry”, a pneumatic elbow exoskeleton that uses force sensors to provide assistive support [39].

**Table 4 sensors-25-04263-t004:** Existing elbow exoskeleton devices.

Name	DoF	Application	Portability	Actuator	Sensor	Ref.
ExoSuit	FE	Assistive	Yes	TSA	Force Sensor, sEMG	[28]
Soft Elbow Exoskeleton	FE	Rehabilitation	Yes	Tendon-Sheath Actuator	sEMG, IMU	[2]
ULIX	FE, SP	Rehabilitation	No	Cable-Driven Actuator (Bowden Cable)	Position Sensor	[3]
Soft Exoskeleton	FE	Rehabilitation	Yes	Cable-Driven Actuator (Bowden Cable)	Force Sensor	[29]
Elbow Exoskeleton	FE	Assistive (Industrial Purposes)	Yes	Series Elastic Actuation	EMG Sensor	[32,33,34]
Elbow-sideWINDER	FE	Assistive (Industrial Purposes)	Yes	Cable-Driven Actuator (Bowden Cable)	Torque Sensor	[50]
Exoskeleton Arm	SP	Assistive	Yes	Cable-Driven Actuator (Bowden Cable)	Non Specified	[51]
Soft Wearable Exosuit	FE	Assistive	Yes	Cable-Driven Actuator (Bowden Cable)	Force Sensor	[30]
Exosuit	FE	Assistive	Yes	Cable-Driven Actuator (Bowden Cable)	IMU	[6]
NEUROExos	FE, SP	Rehabilitation	No	Electric Motor	Position, Torque Sensor	[52]
Elbow Exoskeleton	FE, SP	Rehabilitation	Yes	SMA	None Specified	[49]
Soft Elbow Exoskeleton	FE, SP	Assistive	No	Motor Tendon Actuator	Infrared Sensor	[31]
Carry	FE	Assistive (Industrial Purposes)	Yes	Pneumatic	Force Sensor	[39]
Soft Robotic Elbow Sleeve	FE	Assistive	Yes	Pneumatic	Force Sensor	[19]
Elbow Exoskeleton	FE	Experimental Purposes	No	TSA	Non Specified	[35]
Elbow Exoskeleton	FE, SP	Rehabilitation	No	VSA	Angle Sensor	[53]
DMLS-VSA Elbow Exoskeleton	FE	Rehabilitation	No	VSA	Force Sensor	[54]

Notes: FE = flexion–extension, SP = supination–pronation.

Introduced in 2020, the ExoSuit by Hosseini et al. [28] is a fully soft, assistive wearable device designed for flexion–extension tasks. It utilizes a TSA, in which an electric motor twists a string to generate linear tension. This compact and lightweight mechanism eliminates the need for bulky gears or pulleys, resulting in a low mechanical impedance system that reduces resistance and enhances user comfort. The ExoSuit is particularly well-suited for daily assistive use due to its modularity and gearless simplicity. It integrates surface electromyography (sEMG) sensors that monitor muscle activity in both the biceps and triceps, enabling adaptive, muscle-effort-sensitive control during both load application and removal. A similar TSA-based design was introduced earlier in 2013 by Popov et al. [35], featuring a rigid two-link structure mimicking human upper and forearm anatomy. This earlier design also benefits from the compactness and efficiency of the TSA, with the actuator connected via a coaxial pulley aligned with the elbow joint. However, its rigid structural materials may limit flexibility and long-term user comfort when compared to the fully soft construction of the ExoSuit.

The soft elbow exoskeleton, developed by Wu et al. [2] in 2023, is a soft-suit system designed for rehabilitation training, particularly targeting flexion–extension movements. It employs compliant tendon-sheath actuators driven by integrated servo motors, with real-time control implemented via a Matlab/xPC platform. The soft structure eliminates issues of joint misalignment, enhancing human–robot coordination and making the device comfortable and suitable for extended use. The system integrates sEMG sensors to monitor muscle activity in the biceps and triceps, along with IMUs to track motion, enabling intention-based adaptive assistance. Fastened with Velcro straps for a secure and wearable fit, this sensor-driven, soft-material design is particularly well-suited for long-term rehabilitation applications.

The Ultralow Impedance Exoskeleton (ULIX) [3], developed by Chen et al. in 2019, is a semi-rigid rehabilitation device designed to assist with elbow flexion–extension and forearm supination–pronation. It features a cable-driven differential transmission paired with series elastic actuators (SEAs), enabling high torque output while keeping the total worn mass under 0.9 kg. By decoupling the actuation hardware from the user via Bowden cable transmissions, the system significantly reduces inertia and enhances user comfort during rehabilitation tasks. The SEAs incorporate nonlinear torsional springs that provide low impedance during free movement and high stiffness under load, allowing for precise and responsive torque control. To further improve control fidelity, ULIX integrates dual encoders in each SEA to measure spring deflection and torque, along with an additional encoder on the forearm ring to compensate for control delays caused by cable slack and stiction. This dual-degree-of-freedom, sensor-rich design makes ULIX particularly well-suited for comprehensive upper-limb rehabilitation.

Several soft exoskeletons such as Soft Exoskeleton by Wei et al. (Figure 3a), Soft Wearable Exosuit by Xiloyannis et al. (Figure 3b), and Lightweight Exosuit by Missiroli et al. (Figure 3e) also utilize Bowden cable transmissions. The designs employ Bowden cable transmissions similar to the ULIX system but differ in their fully wearable and mobile configurations. These devices prioritize flexibility and user mobility by utilizing soft fabric structures, thereby enhancing comfort and wearability. In the 2018 design, designed by Wei et al., the motor is mounted remotely to minimize bulk at the joint, with force transmitted through Bowden cables [29]. Ergonomic considerations are central to the system, with carefully positioned anchor points on the textile interface and Velcro straps ensuring a secure yet comfortable fit. These exoskeletons are typically classified as soft elbow devices and vary in their sensor setups; some use force sensors, while others incorporate IMUs for motion detection enabling different modes of assistive support depending on the application [6,29,30].

Among EMG-responsive systems, the Elbow Exoskeleton (2023) by Emir Mobedi highlights the effectiveness of surface EMG sensors in detecting user intention. This device also focuses on improving ergonomics in industrial tasks involving repetitive and high-effort movements, building on prior work demonstrating the value of EMG-based control. It features a lightweight, cable-driven actuator using a series elastic actuation (SEA) system with a bungee element that mimics the damping and compliance of human muscles. Designed to support a single degree of freedom (elbow flexion/extension), the system incorporates a custom spool mechanism, an encoder for joint position tracking, a load cell for cable tension measurement, and an EMG interface for adaptive force control based on muscle activity [32,33,34].

The Elbow-sideWINDER, developed by Park et al. [50], is a semi-rigid exoskeleton designed to support elbow flexion–extension during dynamic tasks in industrial environments, particularly those involving confined spaces. It employs a tendon-driven actuation system using Bowden cables, which enables compact force transmission while minimizing inertia and bulk. Control is achieved through a Myo armband that integrates eight sEMG sensors and a three-axis accelerometer. These sensors estimate external loads based on muscle activity and track arm kinematics, respectively, feeding into a medium-level controller that dynamically adjusts assistive torque in real time. This streamlined, sensor-driven approach allows for effective load compensation with minimal hardware complexity, making the Elbow-sideWINDER well-suited for practical, real-world ergonomic applications [50].

The Exoskeleton Arm, developed by Dezman et al. in 2023, is a rigid, modular device designed specifically to support forearm supination–pronation in rehabilitation contexts. It employs Bowden cable actuation and introduces a novel guided-rod mechanism in place of traditional C-ring bearings, enabling efficient rotational movement while reducing overall volume and weight. This design allows the system to be used independently or as a supplementary module alongside flexion–extension exoskeletons. While Bowden cable slack and stretch remain technical challenges, the device’s modularity makes it particularly well-suited for targeted rehabilitation where supination–pronation recovery is essential [51].

The NEUROExos, developed by Vitiello et al., is a rigid elbow exoskeleton designed for neurological rehabilitation, particularly post-stroke recovery, offering one degree of freedom focused on elbow flexion and extension. It employs a custom-designed smart series elastic actuator (SEA) that enables compliant and precise force control, critical for safe physical human–robot interaction. The SEA design facilitates impedance and admittance control strategies, allowing the device to adapt to patient-specific needs during therapy. The system integrates various sensors, including torque sensors within the actuator, position sensors (encoders), and force sensors, which together enable accurate joint torque measurement, motion tracking, and real-time monitoring of user–robot interaction forces. These features support adaptive therapy modes and ensure both safety and effectiveness in upper-limb rehabilitation [52].

Upper limb rehabilitation exoskeletons designed by Copaco et al. (Figure 3d) use special SMA actuators that are light (less than 1 kg) and quiet, making them comfortable to wear. These actuators work in pairs to control the elbow’s bending (flexion–extension) and rotation (supination–pronation), which means the exoskeleton can move in multiple ways like a natural arm. The design is simple and adjustable, so patients can easily put it on, and it is made with low-cost parts including 3D-printed pieces. The device can either help move the arm (active mode) or just record movement data while the patient moves on their own (passive mode). Sensors inside the exoskeleton track the arm’s position accurately, helping the actuators work smoothly. This combination of sensors, actuators, and multiple movement options makes the exoskeleton a useful and affordable tool for arm rehabilitation [49].

A range of innovative elbow exoskeletons has emerged in recent years, exploring both motor-tendon and pneumatic actuation methods for assistive and rehabilitation purposes. The Soft Elbow Exoskeleton (Figure 3c) features a dual-degree-of-freedom motor-tendon actuator system combined with infrared sensors [31]. In the pneumatic category, designs like the Robotic Elbow Sleeve [19] and the Carry exoskeleton (Figure 3f) provide high compliance and safe force delivery through air-powered actuators enclosed in textile sleeves. Specifically, the Carry system is designed for industrial and assistive tasks such as load holding and carrying, delivering up to 11 Nm of torque at 3.65 bar. Pressure sensors operating at 100 Hz enable responsive torque control, while the system demonstrates reductions in EMG activity, metabolic cost, and user fatigue. However, all pneumatic systems face limitations in portability and dynamic responsiveness due to reliance on external compressors, actuator filling/release cycles, and pressure constraints [39].

Recently, VSAs have become increasingly popular in the design of elbow exoskeletons, aiming to enhance rehabilitation by adapting to the complex and variable biomechanics of the human elbow. Two notable designs exemplify this trend: Yang et al. [53] developed a cable-driven elbow exoskeleton featuring a symmetric dual-motor VSA combined with a deviation compensation unit to accommodate the changing elbow stiffness and the shifting rotation axis during movement. Their system enables real-time stiffness adjustment and accurate alignment with the anatomical joint, improving comfort and safety in therapy. Meanwhile, Zhang et al. [54] introduced a novel dual-motor load-sharing VSA (DMLS-VSA) employing a planetary gear-based variable stiffness mechanism that allows both motors to simultaneously contribute to power output. This design achieves a broader and more precise stiffness range with faster response times through an innovative cascade PI control scheme, enhancing compliance and adaptability during human–robot interaction. Together, these advances reflect a shift toward more sophisticated, responsive, and anatomically aware VSA implementations in elbow exoskeletons, striving to improve rehabilitation effectiveness and patient safety.

While Table 4 focuses solely on elbow exoskeletons, there are also upper-limb exoskeletons designed to accommodate a wider range of degrees of freedom (DOFs), including shoulder and hand movements. In 2020, Yongtae et al. [36] developed an upper-limb exosuit intended for both industrial and rehabilitation applications. This exosuit is equipped with EMG sensors, a cable-driven actuation system using Bowden cables, and a voice recognition interface.

In 2021, Gull et al. introduced a four-DOF upper-limb exoskeleton designed to assist individuals with disabilities. This system aimed to provide effective motion support across multiple joints [55]. In 2022, Paterna et al. proposed a passive upper-limb exoskeleton utilizing pneumatic artificial muscles, specifically targeting industrial use cases to reduce physical strain during repetitive tasks [37].

Various actuation mechanisms have also been explored in recent designs. The Harmony exoskeleton, for instance, employs series elastic actuators (SEAs) and offers seven DOFs—comprising five active DOFs for the shoulder, one for the elbow, and one for the wrist—providing comprehensive upper-limb assistance [56]. In the same year, Park et al. developed a soft exosuit that uses shape memory alloys (SMAs) as actuators, offering a lightweight and compliant solution for upper-limb support [4].

In 2018, Liu et al. [44] presented a portable variable-stiffness elbow exoskeleton (PVSED) for home-based post-stroke rehabilitation. The device features six degrees of freedom (DOFs): three passive DOFs for human motion, two for adjustable frame length, and one active DOF at the elbow actuated by a motor with an integrated VSA. The VSA adjusts stiffness by shifting a pivot point, enabling customizable assistance based on impairment level. For precise closed-loop control, the system uses a GY-25 angle sensor (0.01° resolution), a six-axis force sensor (0.04 N resolution), and an MTx inertial sensor (0.05° resolution) to monitor joint movement and interaction force. Experiments showed that increasing joint stiffness reduced angular error, supporting effective assist-as-needed rehabilitation.

Two additional full upper-limb exoskeleton systems worth noting are ETS-MARSE and the EXOTIC Exoskeleton. ETS-MARSE is a seven-DOF exoskeleton designed for rehabilitation purposes, assisting movement across the shoulder, elbow, forearm, and wrist joints [21]. In contrast, the EXOTIC Exoskeleton provides five DOFs and features adjustable link lengths. Thøgersen et al. emphasized user safety in the EXOTIC design by incorporating mechanical stoppers, current limiters, an open-brace configuration, and a control system that only actuates upon user command. Additionally, the device is lightweight and includes a unique tongue-control interface for enhanced accessibility [17].

Elbow exoskeletons vary widely in their mechanical and control strategies. Soft, cable-driven systems dominate wearable, home friendly solutions, while rigid or pneumatic systems serve more specialized or clinical needs. Each system contributes uniquely to the evolving landscape of upper limb rehabilitation technologies.

## 9. Design, Engineering, and Adoption Challenges of Elbow Exoskeleton

Designing an elbow exoskeleton presents a unique blend of technical and practical challenges. While engineers must address complex biomechanical and mechanical constraints to create a device that is functional, comfortable, and safe, equally significant hurdles arise when it comes to real-world adoption. Factors such as user acceptance, cost, regulatory approvals, and workplace integration all influence whether these innovative devices can transition from prototypes to everyday tools. This section explores both the design and engineering obstacles and the adoption challenges of elbow exoskeletons.

### 9.1. Design and Engineering Challenges

The development of elbow exoskeletons involves not only the optimization of mechanical performance and control [55] but also a comprehensive consideration of human factors, usability, and practical deployment. The challenges are multifaceted, spanning actuator mechanics, signal processing, and user-centered design elements such as comfort, adjustability, and appearance.

This section integrates challenges identified across the literature with analysis from current exoskeleton technologies (see Section 8).

#### 9.1.1. Actuator and Responsiveness

A critical engineering challenge lies in designing actuators that deliver sufficient torque for elbow flexion and extension while maintaining a compact, lightweight form factor. Pneumatic actuators, as demonstrated in the soft robotic elbow sleeve by Koh et al. [19], offer safe and compliant motion but require external air compressors, increasing overall system weight and reducing portability. Cable-driven systems, like those used in Elbow-sideWINDER [50] and ULIX [3], address portability by offloading motor mass to remote locations but introduce frictional losses and control complexity. TSA-based systems like ExoSuit [28] offer a compact and cost-effective solution, though they face nonlinear behavior and string fatigue over time [35,57].

#### 9.1.2. Comfort and Wearability

For exoskeletons intended for prolonged use, especially in rehabilitation, comfort is paramount. Rigid or bulky components can cause discomfort, fatigue, or even misalignment with human joints. Designs must minimize contact pressure, use breathable and soft materials, and consider weight distribution. Soft wearable systems such as those by Wu and Chen [2] and Xiloyannis et al. [30] show promise.

#### 9.1.3. Weight and Device Placement

Weight distribution affects both usability and safety. Devices must avoid placing excessive weight on the forearm or shoulder. Poor placement can lead to strain, reduced range of motion, or unnatural joint alignment. In soft robotic designs, there is a push toward decentralizing components (e.g., compressors, batteries) to the waist or back, as seen in TSA- and Bowden-based systems [2,28,50]. This improves ergonomics but might introduce cabling and connection management issues.

#### 9.1.4. Safety and User Protection

Safety is a non-negotiable aspect, especially in rehabilitation. Devices must prevent hyperextension, respond appropriately to involuntary user motion, and fail gracefully in case of malfunction. Similarly, in the Elbow Exoskeleton designed by Ismail et al. [31] and Upper Limb Exoskeleton designed by Thorgesen et al. [17], user safety is a primary consideration during both wear and operation.

#### 9.1.5. Sensor Accuracy and Stability

Effective control systems require stable and accurate sensor inputs. sEMG signals, while promising for intent detection, suffer from signal variability, placement sensitivity, and susceptibility to noise. This challenge requires robust filtering, threshold calibration, and session-specific tuning. Alternatives like joint torque sensors (e.g., in Elbow-sideWINDER [50]) and IMUs (e.g., [2,6]) provide consistent motion data but introduce complexity in mechanical integration and signal drift mitigation.

#### 9.1.6. Cost and Accessibility

Many current exoskeleton prototypes are expensive to produce, limiting accessibility outside research or hospital settings. High costs stem from custom components, high precision actuators, and integrated electronics. Future engineering must focus on low-cost materials, modular assembly, and open-source electronics to enable community rehabilitation and assistive use at scale. Lightweight fabrication techniques like 3D printing and textile-based integration offer promising avenues [28,49].

#### 9.1.7. Adjustability and Fit

One size does not fit all. The customizability of joint alignment, strap tension, and arm dimensions is crucial, especially for systems targeting diverse populations such as elderly individuals or stroke patients. Many designs still lack simple, tool-free adjustment mechanisms. Modular architectures or self-aligning joints, like those proposed in NEUROExos [52], provide a solution, albeit at the cost of added complexity.

#### 9.1.8. Ease of Use and Setup

The devices should be easy to don and doff independently. Primarily to avoid time-consuming setup, especially requiring a therapist’s assistance (rehabilitation application), future systems must aim for plug-and-play usability, minimal calibration, and, ideally, self-guided instructions. Systems like [2] show progress in this area with user-friendly interfaces.

#### 9.1.9. Design Appearance

Appearance may seem secondary but plays a significant role in user adherence, especially for wearables used outside clinical settings. Devices that resemble medical or industrial equipment may deter consistent use due to social stigma or aesthetic discomfort. Incorporating slim profiles, textile coverings, or clothing-like integration (as in soft systems [2,6,30]) may improve adoption and comfort, especially in social or work environments.

### 9.2. Adoption and Real-World Implementation Challenges

Despite the growing body of research on exoskeletons, their adoption in real-world settings remains limited due to a combination of scientific, technical, behavioral, economic, and institutional barriers.

#### 9.2.1. Limited Long-Term Evidence and Field Validation

One of the primary issues is the lack of long-term field studies that demonstrate the sustained effectiveness and safety of exoskeletons in everyday work environments. While laboratory-based research has shown promising results particularly in reducing work-related musculoskeletal disorders (WMSDs), these findings often do not translate easily into complex, real-world contexts. Organizations are hesitant to adopt exoskeletons without robust longitudinal data that confirm their benefits over time and across varied use cases [58,59].

#### 9.2.2. User Acceptability and Behavioral Resistance

User acceptability is another significant barrier. In both rehabilitation and occupational settings, there is concern that exoskeletons may reduce the user’s sense of agency, relegating them to a more passive role. This resistance can be psychological as much as physical, particularly when the technology is perceived as uncomfortable, intrusive, or incompatible with the natural movements of the human body [59,60].

#### 9.2.3. Lack of Stakeholder Integration

Institutionally, the adoption process is often fragmented. Key stakeholders including workers, health and safety personnel, human resources, unions, and policymakers are not always adequately involved in the selection, evaluation, or implementation phases. This disconnect leads to a lack of shared understanding and commitment, which is critical for successful integration [59].

#### 9.2.4. Economic and Organizational Factors

Economic and organizational barriers such as high initial costs, unclear return on investment, and limited reimbursement models contribute to hesitancy, particularly in sectors like healthcare and public services where procurement processes are complex and budget-driven. Compounding these challenges is a lack of policy guidance and public awareness about the broader societal applications of exoskeletons beyond industrial use. While robotics have become more common in manufacturing, surgery, and logistics, their social and rehabilitative applications are still emerging, often without sufficient support from government policies or institutional strategies. This reflects a broader issue where research and technological development are not yet fully aligned with practical implementation needs [60,61].

To enable wider adoption, research must expand beyond technical performance to include user experience, workplace dynamics, policy design, and long-term impact assessments ensuring that the integration of exoskeletons is not only technologically sound but also socially and economically sustainable.

## 10. Future Direction and Trends

As elbow exoskeleton technology advances, new frontiers are emerging that aim to enhance user comfort, functionality, and accessibility across both clinical and daily life settings. Driven by interdisciplinary innovation across robotics, materials science, and neuroscience, the future of elbow exoskeletons will likely be defined by smart adaptability, seamless human–robot interaction, and integration into everyday rehabilitation and assistive routines.

### 10.1. Toward Fully Wearable and Portable System

One of the clearest trends is the miniaturization and portability of components. Future exoskeletons will increasingly abandon stationary or bulky support systems, such as external air compressors used in pneumatic actuators, in favor of self-contained, battery-powered units. Innovations like lightweight fabric-based actuators or twisted string systems have the potential to replace traditional rigid motors and bulky pneumatics while maintaining comparable torque outputs. Similar to several exoskeleton devices that have been developed, such as the soft actuated glove [62] and the MGlove-TS [63], which are exoskeletons for the hand and wrist, TSA has demonstrated good and promising performance, particularly in the field of rehabilitation. The use of TSA in these devices creates a more flexible and lightweight environment compared to other types of actuators [64,65].

### 10.2. Smart and Adaptive

A shift toward intention aware exoskeletons is evident in recent work, such as the soft elbow exoskeleton that uses sEMG-based control to interpret muscle signals for active assistance alongside IMU data [2]. Future systems are expected to leverage machine learning algorithms to classify and adapt to individual movement patterns, fatigue levels, and rehabilitation goals over time. Furthermore, the integration of multimodal sensing such as pressure sensors, torque sensors, and various biosignals will enable context-aware behavior, allowing the exoskeleton to dynamically adjust support levels in real time. This closed-loop, adaptive control approach promises to enhance user comfort and safety while minimizing the need for manual recalibration.

Although these sensors have not yet been implemented in the elbow exoskeletons discussed in this paper (Section 8), recent advances in soft skin sensor technologies present exciting opportunities for future integration. A notable example is the three-axis Hall effect-based sensor developed by Tomo et al. [66], which features a silicone-encased array capable of detecting pressure and shear forces in three dimensions. With its soft, conformal construction, multi-axis sensitivity, and built-in temperature compensation, this sensor is particularly well-suited for providing accurate and comfortable real-time force feedback critical for safe and responsive exoskeleton control.

Additionally, skin-interfaced wearable sensor systems such as those reviewed by Ray et al. [67] offer stretchable and conformal platforms capable of monitoring joint motion, strain, and physiological signals during physical activity. These systems are currently used in fields like sports science and performance monitoring, where continuous real-time feedback is essential. Their potential integration into elbow exoskeleton cuffs could enable seamless biomechanical and physiological data acquisition without compromising comfort or mobility. Recent developments have also introduced soft, stretchable sensors that support in situ sweat analysis and biochemical monitoring, further expanding the possibilities for holistic, real-time feedback. If applied to future exoskeleton designs, these technologies could significantly improve personalization, adaptiveness, and health awareness in wearable assistive devices.

### 10.3. Cost-Effective Manufacturing

Despite the promising clinical results, most elbow exoskeletons are limited to research labs or specialized clinics due to high costs and limited scalability. A future trend will be focused on low-cost, modular designs using 3D printing, open-source control electronics, and readily available materials. Modular exoskeleton kits that can be custom fitted or 3D printed on demand could democratize access, particularly in underserved or remote healthcare settings. The use of low-cost actuators is also a viable solution; in this regard, TSA offer a more cost-effective alternative compared to other actuator types [63].

## 11. Discussion and Conclusions

Elbow exoskeletons have emerged as essential tools in modern rehabilitation and assistive technologies. Their ability to support, augment, or restore arm movement makes them valuable across diverse applications from clinical recovery programs and daily assistive use to industrial support tasks. This review has highlighted the broad spectrum of mechanical structures, actuation systems, sensor integrations, and control strategies that define the current landscape of elbow exoskeleton development.

A prominent trend in recent years is the growing interest in cable-driven actuation mechanisms, particularly Bowden cables and TSA. While Bowden cable systems are widely adopted for their flexibility and remote actuation capabilities, TSA remains relatively underutilized in elbow exoskeletons despite its promising attributes. TSA systems offer a compact, lightweight, and low-impedance alternative that enhances user comfort and could become more prevalent as research into soft, adaptive devices advances.

Another notable observation is that most elbow exoskeletons currently focus exclusively on flexion–extension movement. This is likely due to the mechanical complexity involved in incorporating both flexion–extension and supination–pronation within a single device. Designing systems that accommodate these combined degrees of freedom remains a significant engineering challenge, often requiring trade-offs in size, weight, and usability.

A recurring theme across the reviewed systems is the trade-off between precision and usability. Rigid exoskeletons often achieve superior control accuracy and torque delivery but may compromise comfort, mobility, and long-term wearability. In contrast, soft-material and cable-driven designs enhance ergonomics and user experience but sometimes fall short in delivering the responsiveness or force required for intensive rehabilitation scenarios. Balancing functional performance with human-centered design remains a critical focus for future innovations.

Despite notable progress, technical challenges persist. These include improving the accuracy of user intention detection (particularly via EMG), ensuring reliable power autonomy for untethered use, enhancing the stability of sensor readings, and optimizing actuator efficiency. Furthermore, the variability among users such as anatomical differences, varying degrees of impairment, and personalized therapy needs calls for adaptable, modular solutions that can be fine-tuned to individual users. The integration of machine learning and intelligent control systems offers promising pathways to address these complexities.

In conclusion, elbow exoskeletons are transitioning rapidly from experimental prototypes to practical, user-centric solutions. The convergence of soft robotics, advanced sensing technologies, and personalized control algorithms is driving the field toward more accessible and effective systems. By embracing interdisciplinary collaboration, focusing on affordability, and maintaining a strong commitment to user-centered design, future elbow exoskeletons can achieve seamless human–machine integration. These advancements hold the potential to restore independence, improve rehabilitation outcomes, and significantly enhance the quality of life for individuals with upper limb impairments around the world.

## Figures and Tables

**Figure 1 sensors-25-04263-f001:**
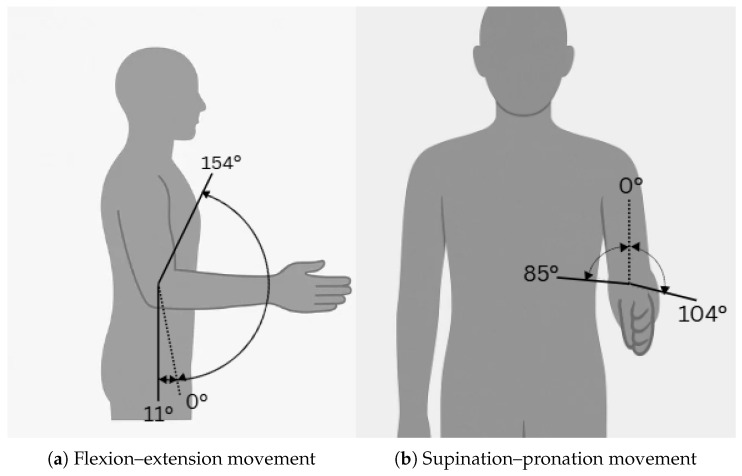
Elbow movement scheme: (**a**) flexion–extension, (**b**) supination–pronation.

**Figure 2 sensors-25-04263-f002:**
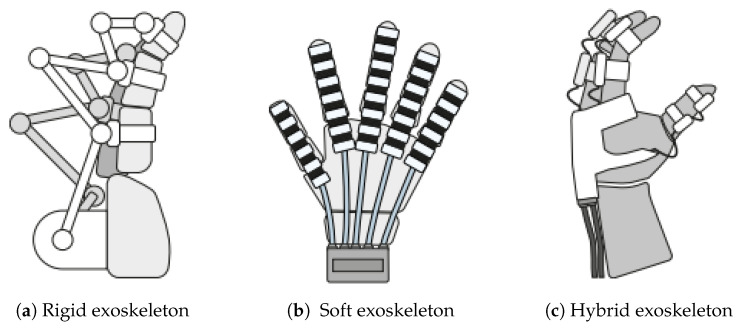
The drawing of three different mechanical structures of exoskeleton: (**a**) rigid exoskeleton, (**b**) soft exoskeleton, and (**c**) hybrid exoskeleton [27].

**Table 1 sensors-25-04263-t001:** Elbow joint range of motion (ROM) and torque.

Motion	ROM (Degrees)	Torque (Nm)	Reference
Flexion	0°–154°	∼2.7	[10,14]
Extension	0°–154°	∼2.7	[10,14]
Pronation	0°–85°	∼0.06	[10,13]
Supination	0°–104°	∼0.06	[10,13]

**Table 2 sensors-25-04263-t002:** Comparison of actuator types used in elbow exoskeletons.

Actuator Type	Power Density	Control Precision	Weight	Portability	Compliance
Pneumatic	Medium	Moderate	Low	Low (requires compressor)	High (air compressibility)
Hydraulic	High	High	High (fluid + hardware)	Low (bulky pump/reservoir)	Low (fluid incompressibility)
Bowden Cable-Driven	Medium	Moderate to High	Low (motor remote)	High	Medium (some compliance)
TSA	Low to Medium	Moderate	Very Low	High	High (soft, compliant)
Electric (Motors)	High	High	Medium to High	High	Low to Medium
SMA	Low	Low to Moderate	Very Low	High	High (material properties)
VSA	Medium to High	High (stiffness and position control)	Medium (due to added stiffness mechanism)	Medium (depends on design)	High (adjustable stiffness improves compliance)

**Table 3 sensors-25-04263-t003:** Comparison of common sensors used in elbow exoskeletons.

Category	Force Sensor	Torque Sensor	EMG Sensor	IMU
Measured Quantity	Linear force	Rotational force (torque)	Muscle activation (bioelectrical signal)	Orientation, acceleration, angular velocity
Advantages	Simple integration, ensures safety	Precise joint control, real-time feedback	Captures user intent, supports voluntary control	Compact, good for motion tracking
Disadvantages	May lack motion detail	Adds bulk, needs careful calibration	Noisy signals, affected by skin/electrode contact	Drift over time, limited accuracy alone
Common Applications	Load monitoring	Joint torque measurement	Intent detection in rehabilitation	Gait and arm motion analysis

## Data Availability

The data presented in this study are available on request from the corresponding author.

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
