# Peer review of "A Comprehensive Review of Elbow Exoskeletons: Classification by Structure, Actuation, and Sensing Technologies"

_sensors, 2025, doi:10.3390/s25144263_

Round 1

Reviewer 1 Report

Comments and Suggestions for Authors

Summary

The article investigates recent development, current designs and engineering trends of elbow exoskeletons by literature review. The authors aim to provide a state-of-the-art overview on elbow exoskeletons and to investigate design challenges and emerging trends to therefore guide the development of elbow exoskeletons. The main scientific contribution is by providing an overview on the current state on elbow exoskeletons with regard to design challenges, corresponding benefits, trade-offs and limitations throughout differing application and providing orientation for potential promising future development of elbow exoskeletons. The review article’s strength lies in the focus on an emerging research field, providing an overview throughout different kinds of application and spreading scientific dissemination over specific fields of research. Another strength lies in the summarization of design and engineering challenges of current elbow exoskeletons, providing guidance for the development of future systems.

General concept comments (The general concept comments and the specific comments are complementary to each other and should therefore be viewed together)

The review is well-structured and provides a comprehensive overview on elbow exoskeletons regarding their technological components, classification, types of application, existing systems, design challenges and trends. Due to expected future development in the field of exoskeletons the research field is relevant. The presented systems and cited references are relevant for the research topic. The presented figures and table are appropriate for visualization, easy to interpret and understandable. The conclusions are drawn coherent and supported by the citations.

The authors are categorizing elbow exoskeletons in active and passive systems stating design differences. The majority of the presented elbow exoskeleton technologies are active system. Multiple text sections are describing exoskeletons as mechatronic systems, indicating active exoskeletons. It remains unclear, if the author’s focus lies upon active elbow exoskeletons. An existing focus should be properly addressed to avoid misinterpretation. Due to differences between active and passive exoskeletons multiple text sections are misleading.

The authors provide information on the anatomy of the human elbow with regard to hard and soft tissue as well as joint structure and movement. This provides necessary information to understand the design specifications and limitations of the elbow exoskeletons. From engineering perspective an abstraction of the complex biomechanical situation seems reasonable. Nevertheless, the biomechanics of joint structure and kinematics of pronation and supination are overly simplified, therefore misleading and need revision.

Multiple different actuation systems for elbow exoskeletons are provided and briefly explained providing information about force and motion generating concepts, helping to understand the exoskeletons movement concepts. Nonetheless not all of the presented actuation systems are represented by the mentioned existing technologies of elbow exoskeleton. Furthermore, there seems to be an overrepresentation of some actuation systems and neglection on others, e. g. hydraulic actuators. This circumstance should be addressed.

The methodology of the literature search is described to ensure a comprehensive review. Google Scholar is mentioned as primary research database for its wide interdisciplinary coverage of research material. Yet no information is given regarding different databases used. Furthermore, information regarding the search strategies, the screening and selection process and the data collection process used are not described in detail. More detailed information about literature search should be giving to improve reproducibility.

Specific comments (The general concept comments and the specific comments are complementary to each other and should therefore be viewed together)

Line 25: Addressing elbow exoskeletons in general as robotic systems can be misleading due to definition of passive elbow exoskeletons.

Line 26: The text section “improving the joint’s mobility” can be misleading, leaving the impression that the natural joint’s range of motion should be increased.

Line 29: For uniformity the term supination-pronation shouldn’t be repeatedly changed to pronation-supination throughout the text.

Line 30: Correction of the word “wearable” seems needed.

Line 30 to 31: There seems to be an issue with the sentence structure, as it is not clear what the authors want to address with it.

Line 36: Addressing elbow exoskeletons in general as mechatronic systems can be misleading due to definition of passive elbow exoskeletons.

Line 41: It is mentioned that exoskeletons aim at supporting the user. The term “resistance” can be misleading since it can indicate an opposing force against the user’s intention.

Line 80: In the mentioned division of the human body the trunk seems missing and should be added.

Line 85: Consistency of abbreviations should be maintained.

Line 85 to 87: The text section indicates that supination-pronation only takes place in the elbow. Regarding biomechanics proximal and distal radioulnar joints are involved in these motions. Therefore, clarification is needed.

Line 87 to 88: For uniformity the term flexion-extension shouldn’t be changed to extension-flexion.

Line 92: It is mentioned that the elbow joint is a complex hinge joint. Due to the involved humerus, radius and ulna and the corresponding joints this statement is a simplification and should be addressed in more detail.

Line 96 to 97: It is mentioned that the radius enables pronation and supination. The joints between humerus, radius and ulna are involved in these motions. Clarification is needed.

Line 103 to 107: The text section can be misleading indicating M. biceps brachii as primary flexor of the elbow.

Line 106: Musculus brachialis spelling mistake.

Line 111 to 116: The authors address the ligaments function by motion pattern of the elbow joint. To assist in the users understanding and to avoid misinterpretation proper terminology, e. g. varus stress, should be used and briefly explained.

Line 119: In the text section 0° elbow flexion is mentioned as full extension. This is contradicted to figure 1 showing an extension of 11°. This should be addressed.

Line 121: The text section indicates that the elbow permits supination-pronation. Regarding biomechanics proximal and distal radioulnar joints are involved in these motions. Therefore, clarification is needed.

Line 168: The authors address, that exoskeletons protect users from both minor and severe injury. The cause of minor and severe injuries as well as kind of protection might want to be addressed to avoid misinterpretation.

Line 172 to 173: It is mentioned that exoskeletons enhance the strength and stamina of soldiers. This statement can be misleading and should be further explained.

Line 196 to 197: Due to the sentence structure it is not fully clear what the authors want to address with it. Furthermore “enhance human movement” can be misinterpreted into increasement of the natural range of motion.

Line 199 to 200: It is not clear what the authors mean by “soft material actuator”. This should be revised.

Line 200: Spelling mistake “system”.

Line 210 to 211: Word “allowing” is separated.

Line 215: Mentioning exoskeletons as systems with an actuator without exclusion of passive systems can be misleading due to the definition of passive exoskeletons.

Line 219 to 220: The authors mention commonly used actuators of elbow exoskeletons. Due to the limited examples of exoskeletons with hydraulic actuator and Shape Memory Alloys mentioned in the existing technologies of elbow exoskeletons it remains questionable if their usage for elbow exoskeletons is common.

Table 1: Misspelling “Pneumatic”.

Line 402: For uniformity the designation rotation shouldn’t be changed to twisting.

Line 611: Misspelling “Polylactide”.

Author Response

General Comment: The review is well-structured and provides a comprehensive overview on elbow exoskeletons regarding their technological components, classification, types of application, existing systems, design challenges and trends. Due to expected future development in the field of exoskeletons the research field is relevant. The presented systems and cited references are relevant for the research topic. The presented figures and table are appropriate for visualization, easy to interpret and understandable. The conclusions are drawn coherent and supported by the citations.

  1. Thank you for the positive feedback on our work and for the insightful comments that helped us to improve it.

The authors are categorizing elbow exoskeletons in active and passive systems stating design differences. The majority of the presented elbow exoskeleton technologies are active system. Multiple text sections are describing exoskeletons as mechatronic systems, indicating active exoskeletons. It remains unclear, if the author’s focus lies upon active elbow exoskeletons. An existing focus should be properly addressed to avoid misinterpretation. Due to differences between active and passive exoskeletons multiple text sections are misleading.

  1. Thank you for the comment. In the revised version, when referring to both active and passive exoskeletons, we replaced the term “mechatronic systems” with “wearable systems” throughout the text to improve clarity and maintain consistency.

The authors provide information on the anatomy of the human elbow with regard to hard and soft tissue as well as joint structure and movement. This provides necessary information to understand the design specifications and limitations of the elbow exoskeletons. From engineering perspective an abstraction of the complex biomechanical situation seems reasonable. Nevertheless, the biomechanics of joint structure and kinematics of pronation and supination are overly simplified, therefore misleading and need revision.

  • Thank you for your comment. In the revised version of the manuscript, we have improved subsection 2.4 by incorporating additional discussion on the biomechanics of the elbow, with a particular focus on pronation and supination. To strengthen this section, we have also included relevant findings from the following study, which we believe provides important insights: 
  • https://www.sciencedirect.com/science/article/pii/S0021929013003527?casa_token=BRR4aCMPcn0AAAAA:Zz69JDacKTHn0eWdJQiodXp-EiJZodaqOO1dp1J1QrM9OQ4iCng6Qe5L6J3EKG4e4UaT74uA

Multiple different actuation systems for elbow exoskeletons are provided and briefly explained providing information about force and motion generating concepts, helping to understand the exoskeletons movement concepts. Nonetheless not all of the presented actuation systems are represented by the mentioned existing technologies of elbow exoskeleton. Furthermore, there seems to be an overrepresentation of some actuation systems and neglection on others, e. g. hydraulic actuators. This circumstance should be addressed.

  • Thank you for pointing this out. We fully agree with your observation, and in the revised version of the manuscript, we have removed the corresponding section as suggested.

The methodology of the literature search is described to ensure a comprehensive review. Google Scholar is mentioned as primary research database for its wide interdisciplinary coverage of research material. Yet no information is given regarding different databases used. Furthermore, information regarding the search strategies, the screening and selection process and the data collection process used are not described in detail. More detailed information about literature search should be giving to improve reproducibility.

  • Thank you for your feedback. We have improved description research methodology by creating a dedicated paper section (new Section 2, entitled Methodology) and by providing by providing more detailed information on the keywords used and including data on the number of papers published per year. We also highlighted trends across specific ranges of years to give a clearer overview of the research landscape. In addition, we have elevated this subsection to a full section to better reflect the expanded content and its importance within the paper. We hope these enhancements address your comment and improve the overall clarity and depth of the manuscript.

Comment R1.1: Line 25: Addressing elbow exoskeletons in general as robotic systems can be misleading due to definition of passive elbow exoskeletons.

  • Thank you for your feedback. In response to your comment, we have revised the terminology throughout the manuscript, changing "robotic systems" to "wearable systems" to more accurately reflect the focus of our work.

Comment R1.2: Line 26: The text section “improving the joint’s mobility” can be misleading, leaving the impression that the natural joint’s range of motion should be increased.

  • Thank you for pointing out the potential ambiguity in the phrase “improving the joint’s mobility.” To avoid any misunderstanding that the natural range of motion should be increased, we have revised the sentence for clarity. The updated text now reads: "These wearable systems are specifically designed to assist users in performing elbow-related movements, supporting joint function and facilitating natural motion. Based on the biomechanics of the human elbow, these systems enable the mechanical transfer of power to produce two primary types of joint motions: flexion-extension and pronation-supination." We believe this revision more accurately reflects the intended meaning and improves the clarity of the manuscript.

Comment R1.3: Line 29: For uniformity the term supination-pronation shouldn’t be repeatedly changed to pronation-supination throughout the text.

  • Thank you for your suggestion regarding terminology consistency. We have updated the manuscript to use the term “supination-pronation” uniformly throughout the text to maintain clarity and consistency.

Comment R1.4: Line 30: Correction of the word “wearable” seems needed.

  • Thank you for noting this. We have corrected the word “wearable” as suggested in the revised manuscript.

Comment R1.5: Line 30 to 31: There seems to be an issue with the sentence structure, as it is not clear what the authors want to address with it.

  • Thank you for your feedback regarding the sentence structure. We have revised the sentence for clarity. The updated sentence now reads: "An exoskeleton is a wearable device designed to support and assist the human body by moving in harmony with its natural motions." We believe this revision clearly conveys the intended meaning.

Comment R1.6: Line 36: Addressing elbow exoskeletons in general as mechatronic systems can be misleading due to definition of passive elbow exoskeletons.

  • Thank you for your comment. To avoid any confusion regarding the term “mechatronic systems,” especially in relation to passive elbow exoskeletons, we have revised the text to refer to these devices more generally as “wearable devices.” We believe this terminology is more inclusive and accurate.

Comment R1.7: Line 41: It is mentioned that exoskeletons aim at supporting the user. The term “resistance” can be misleading since it can indicate an opposing force against the user’s intention.

  • Thank you for your comment regarding the use of the term “resistance.” We acknowledge that this term can be interpreted in different ways depending on the context. To clarify, in assistive applications, exoskeletons provide support to the user’s intended movements, while in rehabilitation settings, exoskeletons based on impedance control can indeed provide controlled resistance to the user’s motion to facilitate therapy. We have revised the manuscript to better reflect this distinction and improve clarity.

Comment R1.8: Line 80: In the mentioned division of the human body the trunk seems missing and should be added.

  • Thank you for your careful reading. We have reviewed the sentence and included the trunk in the division of the human body as suggested.

Comment R1.9: Line 85: Consistency of abbreviations should be maintained.

  • Thank you for your comment. We have carefully reviewed the manuscript and ensured consistency of abbreviations throughout the text.

Comment R1.10: Line 85 to 87: The text section indicates that supination-pronation only takes place in the elbow. Regarding biomechanics proximal and distal radioulnar joints are involved in these motions. Therefore, clarification is needed.

  • Thank you for your insightful comment. We agree that supination-pronation involves both the proximal and distal radioulnar joints, not solely the elbow. In the revised manuscript, we have updated the text to clarify this point. The new wording is as follows: "As noted by Gull and colleagues, most studies model the elbow as having only one degree of freedom, typically for flexion-extension. However, the forearm enables a second rotational movement supination-pronation which involves both the proximal and distal radioulnar joints, though it is often functionally associated with the elbow." We believe this revision improves the anatomical accuracy and clarity of the manuscript.

Comment R1.11: Line 87 to 88: For uniformity the term flexion-extension shouldn’t be changed to extension-flexion.

  • Thank you for your suggestion. We have revised the manuscript to use the term “flexion-extension” consistently throughout the text.

Comment R1.12: Line 92: It is mentioned that the elbow joint is a complex hinge joint. Due to the involved humerus, radius and ulna and the corresponding joints this statement is a simplification and should be addressed in more detail.

  • Thank you for your comment. We agree that describing the elbow joint simply as a “complex hinge joint” is a simplification. In the revised version, we have improved subsection 3.1 to provide a more detailed and accurate description of the elbow joint. We believe this enhancement adds clarity and depth to the manuscript.

Comment R1.13: Line 96 to 97: It is mentioned that the radius enables pronation and supination. The joints between humerus, radius and ulna are involved in these motions. Clarification is needed.

  • Thank you for your comment. In the revised version we Improved subsection 3.1.

Comment R1.14: Line 103 to 107: The text section can be misleading indicating M. biceps brachii as primary flexor of the elbow.

  • Thank you for your observation. To avoid any misleading implication that M. biceps brachii is the primary flexor of the elbow, we have revised the text by changing “primarily responsible” to “responsible.” This adjustment better reflects the contributions of multiple muscles involved in elbow flexion.

Comment R1.15: Line 106: Musculus brachialis spelling mistake.

  • Thank you for catching the spelling mistake. We have corrected the spelling of brachialis in the revised manuscript.

Comment R1.16: Line 111 to 116: The authors address the ligaments function by motion pattern of the elbow joint. To assist in the users understanding and to avoid misinterpretation proper terminology, e. g. varus stress, should be used and briefly explained.

  • Thank you for your suggestion. To enhance clarity and avoid any misinterpretation, we have improved subsection 3.3 in the revised manuscript by incorporating proper terminology, such as “varus stress ”. We believe this will assist readers in better understanding the ligament functions related to the elbow joint’s motion patterns.

Comment R1.17: Line 119: In the text section 0° elbow flexion is mentioned as full extension. This is contradicted to figure 1 showing an extension of 11°. This should be addressed.

  • Thank you for pointing out this inconsistency. We have revised the manuscript to clarify the elbow flexion angles and better explain the range of motion. Specifically, we cite reference in the figure description and include the following explanation: The standard anatomical reference for elbow flexion begins at 0°, representing full extension where the arm is straightened alongside the body. However, as shown in Figure 1, some individuals may exhibit slight hyperextension, up to 11° beyond neutral, due to natural joint variability. The normal range of flexion extends up to approximately 154°, enabling common activities such as lifting and reaching.We believe this clarification resolves the apparent contradiction and improves the manuscript’s accuracy.

Comment R1.18: Line 121: The text section indicates that the elbow permits supination-pronation. Regarding biomechanics proximal and distal radioulnar joints are involved in these motions. Therefore, clarification is needed.

  • Thank you for your comment. We agree that supination-pronation involves both the proximal and distal radioulnar joints. To clarify this, we have improved the relevant text in the manuscript to accurately reflect the biomechanics of these motions.

Comment R1.19: Line 168: The authors address, that exoskeletons protect users from both minor and severe injury. The cause of minor and severe injuries as well as kind of protection might want to be addressed to avoid misinterpretation.

  • Thank you for your suggestion. To avoid any potential misinterpretation, we have expanded the discussion to describe in more detail the causes of both minor and severe injuries and the types of protection that exoskeletons provide. We have also included specific examples to clarify these points in the revised manuscript. The following sources were used to support this statement:
  • https://onepetro.org/ASSPPDCE/proceedings-abstract/ASSE17/All-ASSE17/77236
  • https://www.sciencedirect.com/science/article/pii/S092575352200282X
  • https://onepetro.org/PS/article-abstract/64/03/32/33546/Exoskeletons-Used-as-a-PPE-for-Injury-Prevention

Comment R1.20: Line 172 to 173: It is mentioned that exoskeletons enhance the strength and stamina of soldiers. This statement can be misleading and should be further explained.

  • Thank you for your comment. To avoid any potential misunderstanding, we have revised the statement to better explain the role of exoskeletons in military applications. The updated sentence now reads: "Similarly, in military applications, exoskeletons are designed to assist soldiers by supporting load-bearing tasks and reducing physical strain, which can help them to carry heavier equipment for extended periods and mitigate fatigue." We believe this revision provides a clearer and more accurate description.

The following source was used to support this statement: 

  • https://journals.sagepub.com/doi/abs/10.1177/0018720820957467

Comment R1.21: Line 196 to 197: Due to the sentence structure it is not fully clear what the authors want to address with it. Furthermore “enhance human movement” can be misinterpreted into increasement of the natural range of motion.

  • Thank you for your valuable feedback. To improve clarity and avoid any misinterpretation regarding the natural range of motion, we have revised the sentence as follows: "On the other hand, a hybrid exoskeleton is designed to assist human movement by providing support during activities, while maintaining a balance between flexibility and structural stability." We believe this revision clearly conveys the intended meaning.

Comment R1.22: Line 199 to 200: It is not clear what the authors mean by “soft material actuator”. This should be revised.

  • Thank you for your comment. To clarify the term “soft material actuator,” we have revised the sentence to include a brief explanation and examples: "Unlike fully soft exoskeletons, which use soft materials (e.g., textile-based materials) for maximum comfortability, hybrid exoskeletons usually use hard materials for their structural support and use soft material actuators for their actuation systems." We believe this revision improves clarity and understanding.

Comment R1.23: Line 200: Spelling mistake “system”.

  • Thank you for pointing out the spelling mistake. We have corrected “system” in the revised manuscript.

Comment R1.24: Line 210 to 211: Word “allowing” is separated.

  • Thank you for pointing out the spelling mistake. We have corrected  “allowing” hypenation in the revised manuscript.

Comment R1.25: Line 215: Mentioning exoskeletons as systems with an actuator without exclusion of passive systems can be misleading due to the definition of passive exoskeletons.

  • Thank you for your important observation. To avoid any confusion regarding passive systems, we have revised the sentence to specify active exoskeletons as follows: "In an active elbow exoskeleton, the actuator serves as the core driving element." We believe this clarification improves accuracy and prevents misinterpretation.

Comment R1.26: Line 219 to 220: The authors mention commonly used actuators of elbow exoskeletons. Due to the limited examples of exoskeletons with hydraulic actuator and Shape Memory Alloys mentioned in the existing technologies of elbow exoskeletons it remains questionable if their usage for elbow exoskeletons is common.

  • Thank you for your comment. To more accurately reflect the prevalence of actuator types in elbow exoskeletons, we have revised the sentence as follows: “There are different types of actuators that have been used in exoskeletons actuation: pneumatic, cable-based, electric, and Shape Memory Alloy. Specifically for elbow exoskeletons, hydraulic and Shape Memory Alloy actuators are less frequent.  For the sake of completeness, however, we included them in the description.

Comment R1.27: Table 1: Misspelling “Pneumatic”.

  • Thank you for pointing out the spelling mistake. We have corrected “pneumatic” in the revised manuscript.

Comment R1.28: Line 402: For uniformity the designation rotation shouldn’t be changed to twisting.

  • Thank you for your suggestion. We have revised the manuscript to use the term rotation consistently instead of “twisting” for uniformity.

Comment R1.29: Line 611: Misspelling “Polylactide”.

  • Thank you for pointing out the spelling mistake. We have corrected “Polylactide” in the revised manuscript.

Reviewer 2 Report

Comments and Suggestions for Authors

This paper reviews elbow exoskeletons. While the current version of the manuscript is well-written, several areas require improvement.

 Firstly, the references cited are insufficient. Key actuation technologies relevant to elbow exoskeletons have not been thoroughly analyzed or reported in comparison to existing approaches. The reviewer recommends including references to research on variable stiffness actuator-based exoskeletons, which have been or could potentially be utilized in elbow exoskeletons:

[1] Y. Liu, S. Guo, H. Hirata, H. Ishihara and T. Tamiya, Development of a powered variable-stiffness exoskeleton device for elbow rehabilitation, Biomedical Microdevices, 20(64), 2018.

[2] M. Cestari, D. Sanz-Merodio, J.C. Arevalo, E. Garcia, An adjustable compliant joint for lower-limb exoskeletons, IEEE/ASME Trans. Mechatron. 20 (2), 2015.

[3] L. Liu, et al., Low impedance-guaranteed gain-scheduled GESO for torque-controlled VSA, with application of exoskeleton-assisted sit-to-stand, IEEE/ASME Trans. Mechatron. 26 (4), 2021.

Secondly, as the paper is submitted to the journal Sensors, greater emphasis should be placed on the application of sensors. The reviewer strongly suggests including discussions on technologies such as soft skin sensors for measuring human activities, which represent a promising direction for integration with elbow exoskeletons.

Author Response

Comment R2.1: Firstly, the references cited are insufficient. Key actuation technologies relevant to elbow exoskeletons have not been thoroughly analyzed or reported in comparison to existing approaches. The reviewer recommends including references to research on variable stiffness actuator-based exoskeletons, which have been or could potentially be utilized in elbow exoskeletons:

[1] Y. Liu, S. Guo, H. Hirata, H. Ishihara and T. Tamiya, Development of a powered variable-stiffness exoskeleton device for elbow rehabilitation, Biomedical Microdevices, 20(64), 2018. 

[2] M. Cestari, D. Sanz-Merodio, J.C. Arevalo, E. Garcia, An adjustable compliant joint for lower-limb exoskeletons, IEEE/ASME Trans. Mechatron. 20 (2), 2015.

[3] L. Liu, et al., Low impedance-guaranteed gain-scheduled GESO for torque-controlled VSA, with application of exoskeleton-assisted sit-to-stand, IEEE/ASME Trans. Mechatron. 26 (4), 2021.

  • Thank you for your suggestions and the references provided. We have incorporated the first paper you provided on upper limb exoskeletons to enrich the discussion in Section 8 and the second and third paper to enrich the subchapter 6.6. We believe these additions improve the comprehensiveness of the manuscript regarding actuation technologies for elbow exoskeletons.

We also have identified and added other relevant papers that specifically address VSAs designed for elbow exoskeletons.

For example, we have included:

  • https://www.cambridge.org/core/journals/robotica/article/cabledriven-elbow-exoskeleton-with-variable-stiffness-actuator-for-upper-limb-rehabilitation/1392C1B17CE9CD9F19AEA1CAB2A4C6D0
  • https://ieeexplore.ieee.org/stamp/stamp.jsp?tp=&arnumber=10960283

Comment R2.2: Secondly, as the paper is submitted to the journal Sensors, greater emphasis should be placed on the application of sensors. The reviewer strongly suggests including discussions on technologies such as soft skin sensors for measuring human activities, which represent a promising direction for integration with elbow exoskeletons.

  • Thank you for your insightful suggestion. In response, we have expanded the discussion in Section 10.2 to include the application of sensors, with a particular focus on soft skin sensors for measuring human activities. These sensors represent a promising avenue for integration with elbow exoskeletons.

We have referenced relevant studies such as:

  • https://www.mdpi.com/1424-8220/16/4/491
  • https://www.sciencedirect.com/science/article/pii/S2468451118300680

Reviewer 3 Report

Comments and Suggestions for Authors
  1. General Comments: This paper comprehensively reviews the latest progress in elbow exoskeleton technology, focusing on its mechanical structure, drive mode and sensor technology, and explores design challenges and future trends. This article is mainly divided into seven parts: First, the author explains the relevant technical indicators of the exoskeleton system, including the range of motion and output torque, based on the anatomy of human joints; then the application of exoskeletons is introduced, including daily exercise assistance for people with movement disorders, rehabilitation training for patients, and enhanced applications for industrial scenarios; next the author divides exoskeletons into three categories according to the material type: rigid, flexible and hybrid, and also according to the structural design divides them into two categories: active systems and passive systems; in the fourth part, the drive systems of various exoskeletons are introduced in detail, covering pneumatic, hydraulic, cable, electric, shape memory alloy, etc.; then, the sensor technology used is also mentioned, mainly including force sensors, torque sensors, surface electromyography sensors and inertial measurement units; in the next part, the author introduces the existing exoskeleton devices in detail, including ExoSuit, Soft Elbow Exoskeleton, ULIX and NEUROExos; in the last part, the author puts forward the design and engineering challenges in this field, and looks forward to the future development direction and trend.
  2. Strength:
    (a) This article summarizes the research and development of exoskeleton systems in recent years. Based on application scenarios and actual products, the system’s requirements, technical indicators, research status, research challenges, and future development trends are introduced and analyzed in detail.
    (b) This article systematically classifies existing technologies, conducts in-depth analysis, and finally proposes a forward-looking technical route.
    (c) This article has clear logic and careful thinking, and has guiding significance for the research and development of the exoskeleton system
  3. Weakness:
    (a) On page 2, lines 64-65, it is mentioned that “In total, 50 research papers were selected after screening for relevance, technical depth, and contribution to the understanding of elbow exoskeleton design and development.” It is suggested that the screening criteria should be mentioned. In addition, more recently published papers should be cited as appropriate unless there are good reasons to cite earlier research papers.
    (b) In the fifth part on pages 6-7, the content about actuators only describes the advantages and disadvantages of various actuators through text content. Appropriate data and charts can be added to further compare the advantages and disadvantages of various actuators in detail.
    (c) The content about sensors in the sixth part on page 8 also needs to include actual data and charts to further compare the advantages and disadvantages of various sensors in detail.

(d) In the seventh section, ”Existing Technologies of Elbow Exoskeleton”, it is suggested to add data to further illustrate the extent to which these technologies improve the portability, comfort and safety of the exoskeleton system. In addition, it is supposed to compare the technical indicators of these exoskeleton systems, and analyze their advantages and disadvantages, rather than just listing their DOFs, actuator types, and sensor types through tables.
(e) In the eighth part, ”Design and Engineering Challenges”, the reasons for the existence of challenges and the attempts of current research in this regard should be further explained in detail through papers, theory or data.
(f) There are a lot of blank spaces on some pages of the paper, for example, page 3 and page 9. It is recommended to adjust the format of the paper.

Author Response

Comment R3.1: (a) On page 2, lines 64-65, it is mentioned that “In total, 50 research papers were selected after screening for relevance, technical depth, and contribution to the understanding of elbow exoskeleton design and development.” It is suggested that the screening criteria should be mentioned. In addition, more recently published papers should be cited as appropriate unless there are good reasons to cite earlier research papers.

  • Thank you for your feedback. We have improved it to a new section, by providing more detailed information on the keywords used and including data on the number of papers published per year. We also highlighted trends across specific ranges of years to give a clearer overview of the research landscape. In addition, we have elevated this subsection to a full section to better reflect the expanded content and its importance within the paper. We hope these enhancements address your comment and improve the overall clarity and depth of the manuscript.

Comment R3.2: (b) In the fifth part on pages 6-7, the content about actuators only describes the advantages and disadvantages of various actuators through text content. Appropriate data and charts can be added to further compare the advantages and disadvantages of various actuators in detail.

  • Thank you for your helpful suggestion. In response, we have added a comparison table in the fifth section to visually present the advantages and disadvantages of various actuator types. The table includes key performance metrics such as power density, control precision, weight, portability, and compliance. We believe this addition enhances clarity and allows for a more informative comparison of actuator technologies used in elbow exoskeletons.

Comment R3.3: (c) The content about sensors in the sixth part on page 8 also needs to include actual data and charts to further compare the advantages and disadvantages of various sensors in detail.

  • Thank you for your constructive feedback. In response, we have added a comparison table in the sixth section that presents various types of sensors used in elbow exoskeletons. The table includes key aspects such as measured quantity, advantages, disadvantages, and common applications. We believe this addition improves the clarity and depth of the discussion, allowing for a more comprehensive comparison of sensor technologies.

Comment R3.4: (d) In the seventh section, ”Existing Technologies of Elbow Exoskeleton”, it is suggested to add data to further illustrate the extent to which these technologies improve the portability, comfort and safety of the exoskeleton system. In addition, it is supposed to compare the technical indicators of these exoskeleton systems, and analyze their advantages and disadvantages, rather than just listing their DOFs, actuator types, and sensor types through tables.

  • Thank you for your valuable recommendation. In response, we have improved the existing table in Section 7 by adding new columns that evaluate the portability and application of each elbow exoskeleton system.

Comment R3.5: (e) In the eighth part, ”Design and Engineering Challenges”, the reasons for the existence of challenges and the attempts of current research in this regard should be further explained in detail through papers, theory or data.

  • Thank you for your insightful comment. In response to your suggestion regarding the “Design and Engineering Challenges” section, we have added a more detailed discussion on the barriers to exoskeleton adoption, drawing on findings from several recent studies. These sources provide a comprehensive overview of the technological, ergonomic, economic, and organizational factors that currently limit large-scale implementation, despite extensive research efforts. We referenced the following key papers to support this analysis:

  • https://www.frontiersin.org/journals/public-health/articles/10.3389/fpubh.2023.979225/full
  • https://www.cambridge.org/core/journals/wearable-technologies/article/occupational-exoskeletons-a-roadmap-toward-largescale-adoption-methodology-and-challenges-of-bringing-exoskeletons-to-workplaces/33096E40515BC1CCBBB53B6A4B328E3F
  • https://ieeexplore.ieee.org/abstract/document/6718049?casa_token=tsrwls_rpkwAAAAA:krhZtlzKgnWu0OncdZ_sJNafaUQw6ZahikkMg6306AK-QXejnAxxizmm_RanbBGYFnXN-hA
  • https://iris.unimore.it/handle/11380/1311046

These references help clarify why, despite significant technological advances, the adoption of exoskeletons remains limited. The discussion now better explains the multifaceted challenges ranging from actuator design and comfort to organizational and economic barriers

Comment R3.6: (f) There are a lot of blank spaces on some pages of the paper, for example, page 3 and page 9. It is recommended to adjust the format of the paper.

  •  Thank you for pointing out. we employed the mdpi template to format the paper, in the revised version of the paper we reduced the blank parts. The revised version of the paper is still not in the final format since we highlighted the applied modifications in blue. We will further optimize the format when the paper contents are finalized after the review process. 

Round 2

Reviewer 2 Report

Comments and Suggestions for Authors

I have reviewed the revised version of this manuscript and the authors’ responses to the previous review comments. The authors have thoroughly addressed all concerns raised in the initial review. 

Reviewer 3 Report

Comments and Suggestions for Authors

The authors have made responses to my previous comments point by point, I am satisfied with the revised version.